# Anisotropic Thermally Conductive Perfluoroalkoxy Composite with Low Dielectric Constant Fabricated by Aligning Boron Nitride Nanosheets via Hot Pressing

**DOI:** 10.3390/polym11101638

**Published:** 2019-10-10

**Authors:** Xinru Zhang, Xinzhi Cai, Xiaoyu Xie, Changyu Pu, Xuanzuo Dong, Zeyi Jiang, Ting Gao, Yujie Ren, Jian Hu, Xinxin Zhang

**Affiliations:** 1School of Energy and Environmental Engineering, University of Science and Technology, Beijing 100083, China; xinruzhang@ustb.edu.cn (X.Z.); b20170070@xs.ustb.edu.cn (X.C.); xiexiaoyuustb@163.com (X.X.); nmcfpcy@163.com (C.P.); 15901217705@163.com (X.D.); gn740924@me.ustb.edu.cn (T.G.); xxzhang@ustb.edu.cn (X.Z.); 2Beijing Engineering Research Center of Energy Saving and Environmental Protection, University of Science and Technology, Beijing 100083, China; 3Beijing Key Laboratory for Energy Saving and Emission Reduction of Metallurgical Industry, University of Science and Technology, Beijing 100083, China; 4China Energy Conservation and Environmental Protection Group, Beijing 100082, China; renyujie@cecep.cn; 5China Energy Conservation and Environmental Protection Group, National Machinery United Electric Power (Ningxia) Co., Ltd., Yinchuan 750011, China; huazhonghujian@163.com

**Keywords:** composite, thermal conductivity, dielectric property, boron nitride nanosheets, alignment, hot pressing

## Abstract

Thermal management has become a critical challenge in electronics and portable devices. To address this issue, polymer composites with high thermal conductivity (TC) and low dielectric property are urgently needed. In this work, we fabricated perfluoroalkoxy (PFA) composite with high anisotropic TC and low dielectric constant by aligning boron nitride nanosheets (BNNs) via hot pressing. We characterized the thermal stability, microstructure, in-plane and through-plane TCs, heat dissipation capability, and dielectric property of the composites. The results indicate that the BNNs–PFA composites possessed good thermal stability. When the BNNs content was higher than 10 wt %, the BNNs were well layer aligned in the PFA matrix, and the composites showed obvious anisotropic TC. The in-plane TC and through-plane TCs of 30 wt % BNNs–PFA composite were 4.65 and 1.94 W m^−1^ K^−1^, respectively. By using the composite in thermal management of high-power LED, we found that alignment of BNNs in composite significantly improves the heat dissipation capability of composite. In addition, the composites exhibited a low dielectric property. This study shows that hot pressing is a facile and low-cost method to fabricate bulk composite with anisotropic TC, which has wide applications in electronic packaging.

## 1. Introduction

With the rapid development of miniaturization and integration of microelectronics and portable devices, thermal management has become a critical challenge. New electronic packaging material with high thermal conductivity and low dielectric property is urgently needed [1,2]. Perfluoroalkoxy (PFA), as a high-performance fluoropolymer with superior chemical resistance, high thermal stability and low dielectric constant, has a strong potential in electronic packaging. Unfortunately, pure PFA has a low thermal conductivity (TC) of approximately 0.27 W m^−1^ K^−1^, [3] providing insufficient heat dissipation for effective thermal management. Therefore, improving the TC of PFA while maintaining its low dielectric property has significant implications for its application in electronics and portable devices.

Over the past two decades, extensive studies have demonstrated that incorporating micro/nano ceramic fillers (such as aluminum oxide, boron nitride, aluminum nitride, and silicon carbide) into polymer matrix is a valid method to improve TC of polymer composite while maintaining its low dielectric property [4,5,6,7,8,9,10]. Among these ceramic fillers, the platelet-like boron nitride nanosheets (BNNs) are particularly promising owing to their high TC, low density and excellent insulating property [11,12,13,14,15,16]. BNNs possess highly anisotropic TC, and their in-plane TC (approximately 600 W m^−1^ K^−1^) is much higher than the through-plane TC (1~10 W m^−1^ K^−1^) [17].

To make full use of this high in-plane TC of BNNs, significant efforts have been devoted to enhancing the composite’s TC by aligning BNNs in polymer matrix [18]. Several methods to fabricate the composite with aligned BNNs have been proposed, such as unidirectional freeze casting, exerting an electric/magnetic field, vacuum filtration, and electrospinning [8,11,12,19,20,21]. For example, on the basis of freeze casting method, Hu et al. fabricated the ordered BN network–epoxy composite with a high TC of 4.42 W m^−1^ K^−1^ at the BN content of 34 vol % [21]. By exerting an external magnetic field, Kim et al. aligned BNNs in polyurethane acrylate composite via depositing the titanium oxide nanoparticles onto BNNs, and reported that for a BNNs content of 30 vol %, the TC of the composite was 1.75 W·m^−1^ K^−1^ along the aligned direction of BNNs [22]. On the basis of the vacuum filtration method, Zhang et al. fabricated layer aligned BNNs-polyvinyl alcohol composite with a through-plane and in-plane TC of 1.63 and 8.44 W m^−1^ K^−1^, respectively, at the BNNs content of 27 vol % [23]. Using the electrospinning method, Chen et al. fabricated the polydimethylsiloxane composite with aligned and interconnected BNNs, and reported that the composite not only exhibited a high TC of 1.94 W m^−1^·K^−1^ along the aligned direction of BNNs but also showed excellent electrical insulating property, at the BNNs content of 15.6 vol % [24].

Recently, on the basis of a facile hot-pressing method, Sun et al. fabricated the polycarbonate composite films with layer-aligned BNNs via pre-coating followed by hot-pressing. They found that the composite film with 18.5 vol % BNNs possessed an in-plane TC of 3.09 W·m^−1^·K^−1^ [25]. Meanwhile, Yu et al. fabricated the composite film with layer-aligned BNNs in thermoplastic polyurethane via solution ball-milling followed by hot-pressing. They reported TCs of 50.3 and 6.9 W·m^−1^ K^−1^ along the in-plane and through-plane directions, respectively, with 95 wt % layer-aligned BNNs [26]. These studies indicate that it is promising to fabricate polymer composites with well aligned BNNs by using this facile and low-cost, hot-pressing method. However, these examples are two-dimensional (2D) films, whereas, in various electronic packaging applications, the 3D bulk composite are more desired.

In this work, we fabricated 3D bulk PFA composites with highly layer-aligned BNNs using hot-pressing. By characterizing the thermal stability, microstructure, in-plane and through-plane TC, heat dissipation capability, as well as dielectric property, we found that the PFA composites possessed highly aligned BNNs, and exhibited excellent thermal stability, high anisotropic TC, and low dielectric property. This study provides a facile and low-cost strategy for fabricating PFA composite with aligned BNNs that can be used in various electronic packaging applications.

## 2. Materials and Methods

### 2.1. Preparation and Characterization of Boron Nitride Nanosheets (BNNs)

The BNNs were prepared by liquid phase exfoliation method according to our previous work [27]. As seen in Appendix A, first, 5 g BN bulk material (101419, CAS: 7440-42-8, purity: 99%, XFNANO Materials Tech Co. Ltd., Nanjing, China) was added into a 500 mL ethanol-water mixture (ethanol: 65 vol %), as seen in Appendix A, and shear mixed using a high-shear laboratory mixer (FA25 FLUKO Equipment Shanghai Co., Ltd., Shanghai, China) at a speed of 10,000 rpm for 30 min. Then, the BN–ethanol-water mixtures were sonicated using a tip sonicator (Scientz-950E, Scientz Biotechnology Co. Ltd., Ningbo, China) at 300 W for 180 min. Afterwards, the obtained dispersions stood over night and the supernatants were centrifuged at 2000 rpm for 30 min (TGL10C, ShangHai Anting Scientific Instrument Factory, Shanghai, China) to sediment the aggregated BN. Finally, the produced BNNs were collected.

The morphology, thickness, and crystal structure of the produced BNNs were characterized. Specifically, the morphology of BNNs was observed using a scanning electron microscopy (SEM) (Nova NanoSEM 430, FEI, Hillsboro, OR, USA) operating at a 5-kV acceleration voltage, by pipetting the BNNs dispersions onto a Si substrate. The thickness of BNNs was examined using transmission electron microscopy (TEM) and Raman spectroscopy. In TEM observation, the BNNs dispersion was pipetted onto carbon-coated copper grids, then observed using a TEM (Tecnai F30, FEI, Hillsboro, OR, USA) at a 200-kV acceleration voltage. In Raman spectroscopy measurement, the BNNs powder was deposited onto a glass slide, then examined using a Raman spectroscopy (LabRAM HR800, Horiba Jobin-Yvon, Montpellier, France) with a 514 nm wavelength laser. The crystal structure of BNNs was determined by the high-resolution TEM (Tecnai F30, FEI, Hillsboro, OR, USA) and electron diffraction pattern.

### 2.2. Fabricating Perfluoroalkoxy (PFA) Composites with Aligned BNNs

Figure 1 shows the method to fabricate the aligned BNNs–PFA composites via solvent-assisted blending followed by hot-pressing. First, the produced BNNs were dispersed in ethanol via sonication. Then, PFA powder (obtained from DuPont, Wilmington, DE, USA) was added into the BNNs dispersions and mixed while removing ethanol with a heating magnetic stirrer for 6 h at 60 °C. Afterwards, the BNNs–PFA slurry was dried at 120 °C for 24 h in a vacuum oven to remove the residual ethanol. The obtained BNNs–PFA powder was put into a mold and heated at 400 °C for 1 h. The molten BNNs–PFA sample was hot-pressed at 380 °C under a pressure of 15 MPa for 15 min, then cooled to room temperature under the ambient pressure to form bulk composites [28]. The properties of these composites were then examined. We performed the above steps for PFA composites with BNNs contents of 1, 5, 10, 15, 20, 25, and 30 wt %.

### 2.3. Characterization of BNNs–PFA Composites

We characterized the thermal stability, microstructure, through-plane and in-plane TCs, the heat dissipation capability, dielectric constant and dielectric loss tangent of the BNNs–PFA composites. The thermal stability of composite was determined by thermogravimetric analysis via using a thermal analysis platform (Labsys Evo, Setaram, Caluire-et-Cuire, France) at a heating rate of 10 °C /min from 25 to 900 °C under a nitrogen atmosphere. The microstructure of composite was observed by an SEM (Nova NanoSEM 430, FEI, Hillsboro, OR, USA). Prior to SEM observation, the composites were fractured in liquid nitrogen to form a neat cross section. The alignment degree of BNNs in PFA matrix was quantitatively evaluated by the parameter of <cos^2^*θ*> [29,30].
(1)$$<cos^{2}\theta >=\frac{\int^{\;}\rho \left(\theta \right)\;\cos^{2}\theta \;\sin\theta \;d\theta }{\int^{\;}\rho \left(\theta \right)\;\sin\theta \;d\theta }$$
where *θ* is the acute angle between the base plane of BNNs and in-plane direction of composites, as seen in Appendix A, and is determined by analyzing the orientation of BNNs in the SEM images of the fractured surfaces of the composites. *ρ(θ*) is the distribution function of *θ*, which can be fitted by,
(2)$$\rho \left(\theta \right)=A_{1}e^{\left(-\theta /t_{1}\right)}+y_{0}$$
where *A*_1_, *t*_1_ and *y*_0_ are the fitting parameters, as seen in Appendix A.

The in-plane TC (i.e., *λ*_‖_) and through-plane TC (i.e., *λ*_⊥_) of the BNNs–PFA composites were determined by *λ* = *α* × *c_p_* × *ρ*, where α is the thermal diffusivity, *c_p_* is the specific heat, and *ρ* is the density of composite sheet. The values of *α* and *c_p_* of the composites were measured using a laser flash thermal analyzer (LFA 427, Netzsch, Bobingen, Germany), and *ρ* was determined by measuring the weight and dimensions of the samples.

To evaluate the application potential of the fabricated BNNs–PFA composites, we measured their heat dissipation capability when attached to a high-power light-emitting diode (LED). Specifically, the back surface of a LED lamp was attached to a composite (diameter: 30 mm; height: 1 mm), and a thin copper sheet was connected to the bottom side of the composite. After the LED was switched on, the temperature of LED was measured using an infrared thermal imaging instrument (E60, FLIR Systems, Portland, OR, USA). We compared the heat dissipation capability of PFA composite with 30 wt % layer aligned BNNs with that of pure PFA.

In addition, the dielectric constant and dielectric loss tangent of the BNNs–PFA composites were measured by a broadband dielectric spectrometer (Concept 80, Novocontrol Technology Company, Montabaur, Germany) under the frequency ranging from 10^2^ to 10^6^ Hz.

### 2.4. Predicting the Composite’s Thermal Conductivity (TC) Using a Modified Effective Medium Theory (EMT) Model

The Effective Medium Theory (EMT) model has been successfully used to predict the composite’s TC by considering the size and orientation of fillers as well as the thermal resistance between fillers and polymer matrix. In this work, we used a modified EMT model to simulate *λ*_‖_ of the BNNs-PFA composites, which is expressed as,
(3)$$\lambda_{\|}=\lambda_{m}\frac{2+f\left[\beta_{1}\left(1-L_{1}\right)\left(1+<cos^{2}\theta >\right)+\beta_{3}\left(1-L_{3}\right)\left(1-<cos^{2}\theta >\right)\right]}{2-f\left[2\beta_{1}L_{1}\left(1+<cos^{2}\theta >\right)+\beta_{3}L_{3}\left(1-<cos^{2}\theta >\right)\right]}$$
(4)$$\mathrm{with}\;\beta_{i}=\frac{\lambda_{i}^{c}-\lambda_{m}}{\lambda_{m}+L_{i}\left(\lambda_{i}^{c}-\lambda_{m}\right)}$$
where λ*_m_* is the TC of matrix, *f* is the volume fraction of BNNs in composite. *L_i_* (*i* = 1, 2, 3) is the geometrical factor and expressed as,
(5)$$L_{1}=L_{2}=\frac{\pi \left(a_{3}/a_{1}\right)}{4},\;L_{3}=1-2L_{1}$$
$\lambda_{i}^{c}$ is the TC of a composite unit cell consisting of BNNs and the interfacial thermal barrier,
(6)$$\lambda_{i}^{c}=\frac{\lambda_{i}}{1+\frac{a_{k}\lambda_{i}}{a_{i}\lambda_{m}}}$$
(7)$$\mathrm{with}\;a_{k}=\lambda_{m}R_{k}$$
where λ*_i_* is TC of BNNs along the *a_i_* (*i* = 1, 2, 3) direction, *a_k_* is Kapitza radius, *R_k_* is interface thermal resistance between BNNs and PFA matrix. TC, size and thickness of BNNs are shown in Appendix A.

## 3. Results and Discussion

### 3.1. Properties of the BNNs Produced by Liquid Phase Exfoliation

The morphology, thickness, and crystal structure of the BNNs were characterized. As shown in the SEM image in Figure 2a, the BNNs had a size of about 1–2 μm. Figure 2b depicts the Raman spectra of the BN bulk material and the produced BNNs, indicating that BN bulk material exhibited a characteristic peak at ~1366 cm^−1^ (which corresponds to the E_2g_ vibrational mode), and the produced BNNs were blue-shifted in comparison with the bulk BN. The above differences in the characteristic peaks between bulk BN and the BNNs suggested that the produced BNNs were minimally layered. Figure 2c depicts the TEM images of the produced BNNs. Figure 2d presents the raw high resolution TEM image of the edge of a BNNs. The results show that the BNNs were minimally layered, with a thickness of approximately 5.6 nm. The results were consistent with the Raman spectra measurement shown in Figure 2b.

The high-resolution TEM images of the BNNs are shown in Figure 2e. The lattice spacing of approximately 0.25 nm suggests that there was no structural distortion for the BNNs [31]. Figure 2f depicts the electron diffraction pattern of the BNNs, which demonstrates that the BNNs maintained a six-fold symmetry pattern, which was a typical crystal structure of BN. Together, the characterization of the morphology, thickness, and crystal structure of BNNs suggested that they were minimally layered and had high structural quality.

### 3.2. Thermal Stability of BNNs–PFA Composite

The thermogravimetric analysis curves were used to evaluate the thermal stability of BNNs–PFA composite. Figure 3a indicates that pure PFA and the BNNs–PFA composite all presented a similar one-step decomposition between 500 and 620 °C, which suggests that the addition of BNNs had little influence on the decomposition behavior of PFA matrix. Figure 3b shows the relationship between the residual content obtained in thermogravimetric analysis and the initial designed BNNs content. Because the BNNs could be stable at 800 °C in nitrogen, the residual content could be used to estimate the BNNs content in PFA matrix. The results demonstrate that the residual content agreed well with the initial designed BNNs content, which suggests that BNNs dispersed well in PFA matrix.

The heat-resistance index (THRI) was used to analyze the thermal stability of the composite (Appendix A) [32]. As shown in Figure 3c, the THRI value for pure PFA was about 275.2 °C, and the values for all the BNNs–PFA composites was slightly lower, ranging from 268.2 to 272.7 °C. This result indicates that the addition of BNNs slightly increases the decomposition of PFA composite. This could be due to the ultrahigh TC of BNNs, which would increase the heat conduction, and accelerate the decomposition of the polymer matrix. [33] In addition, the addition of BNNs to the PFA may also reduce the thermal stability by increasing the number of defects in the composite [34].

### 3.3. Microstructure of the BNNs–PFA Composites

Figure 4 shows the SEM images of the fractured surfaces for pure PFA and the BNNs–PFA composites. The image in Figure 4a shows pure PFA matrix had a smooth surface. Figure 4(b1) depicts the SEM image of 1 wt % BNNs–PFA composite, and Figure 4(b2) shows the enlarged view outlined in the orange rectangle area in Figure 4(b1). The results indicate that only a small number of BNNs can be found in the observation. As seen in Figure 4(c1,c2), in the composites with 10 wt % BNNs, the BNNs were well dispersed in the PFA matrix, and a small number of BNNs was aligned in PFA composites; however, few BNNs contacted each other. As seen in Figure 4(d1,d2,e1,e2), at higher BNNs contents of 20 and 30 wt %, more BNNs were layer aligned in PFA composites, and a higher number of BNNs contacted each other in comparison with at 10 wt % BNNs. Evidently, this contact and overlapping of the aligned BNNs would lead to the formation of thermally conductive paths in PFA composite, which is one of the dominant factors to improve the TC of the composites.

To quantitatively evaluate the alignment degree of BNNs in PFA composites, the frequency of *θ* for the BNNs–PFA composites (BNNs > 10 wt %) was determined. Appendix A shows the frequency versus *θ* obtained from the SEM images of PFA composites containing 10, 15, 20, 25, and 30 wt % BNNs. Then, the <cos^2^*θ*> was calculated based on Equations (1) and (2). When <cos^2^*θ*> equals 1/3, BNNs can be considered as randomly dispersed in polymer matrix; in contrast, <cos^2^*θ*> with a value of 1 suggests completely layer aligned BNNs in polymer [30,35]. Figure 4f shows the determined <cos^2^*θ*> for the PFA composites containing 10, 15, 20, 25, and 30 wt % BNNs. The <cos^2^*θ*> for these five composites were all higher than 1/3, and increased with the increasing BNNs content, indicating an increasing alignment of BNNs in composites. In particular, the <cos^2^*θ*> of the 10 wt % BNNs–PFA composite was approximately 0.58, and the values for the PFA composite containing 15, 20, and 25 wt % BNNs were approximately 0.72. In comparison, the <cos^2^*θ*> of the 30 wt % BNNs–PFA composite increased to 0.83 (close to 1), suggesting that most of BNNs were layer aligned. This may be due to the increased compactness of BNNs in polymer matrix [36]. In summary, the above results demonstrated that, when the BNNs content exceeded 10 wt %, the BNNs were well layer-aligned in PFA matrix.

### 3.4. The TC and Heat Dissipation Capability of BNNs–PFA Composites

Figure 5a shows the in-plane (*λ*_‖_) and through-plane (*λ*_⊥_) TCs of pure PFA and the BNNs–PFA composites. The results indicated that pure PFA had a low TC of 0.26 W m^−1^ K^−1^, consistent with the previous study [3]. When the BNNs content exceeded 5 wt %, the *λ*_‖_ was higher than their *λ*_⊥_. When the BNNs content was higher than 10 wt %, the *λ*_‖_ showed a sudden increase compared with their λ_⊥_. In particular, when the BNNs content was 30 wt %, the *λ*_‖_ and *λ*_⊥_ of BNNs–PFA composite was 4.65 and 1.94 W·m^−1^ K^−1^, respectively. The results demonstrated that the BNNs–PFA composites (BNNs content > 10 wt %) fabricated by hot-pressing method showed obvious anisotropic TC.

To analyze the enhanced *λ*_‖_, we used an EMT model to simulate *λ*_‖_ of the BNNs–PFA composites with different *R_k_* and the determined <cos^2^*θ*>. As shown in Figure 5b, the *λ*_‖_ obtained by the EMT model via the trial and error study agreed well with the *λ*_‖_ measured in experiment. In addition, we found that *R_k_* decreased with the increasing BNNs content. In particular, when the BNNs content was 10 wt%, the *R_k_* was 15 × 10^−8^ m^2^·K·W^−1^. In contrast, as the BNNs content increased to 30 wt%, the *R_k_* decreased to 11 × 10^−8^ m^2^·K·W^−1^. The decrease of *R_k_* with the increasing BNNs content may be due to the contact and overlapping of the layer aligned BNNs, and the resulting formation of the thermally conductive paths in PFA matrix, which could reduce the *R_k_* effectively.

Furthermore, we compared the heat dissipation capability of the 30 wt % BNNs–PFA composite with that of pure PFA, by using them in the thermal management of high-power LED. Figure 6a shows the infrared thermal images for LED lamps attached to pure PFA and PFA composite with 30 wt % layer-aligned BNNs, after switching the LEDs on for 30, 120, and 300 s. The surface temperature of LED attached to the 30 wt % BNNs–PFA composite was much lower than that attached to pure PFA. The reason for the higher heat-dissipation capability of this composite than pure PFA is shown in Figure 6b. In terms of the PFA composite containing layer aligned BNNs, although the heat engendered by LED is supposed to transfer to the composite along the Z axis then to the copper sheet, the lower TC of composite along Z axis makes a part of heat flow transfers along the X-Y plane, then to the environment. [37] The heat change with the copper sheet and environment would result in a higher heat dissipation capability of PFA composite with layer aligned BNNs.

Figure 6c shows the surface temperatures of LED lamps as a function of time. The surface temperature of LED attached to pure PFA increased to about 65 °C within 150 s after it was switched on. By contrast, the temperature of LED attached to 30 wt % BNNs–PFA composite increased to approximately 52 °C within 60 s, and remained stable at this temperature. Thus, the temperature of the LED attached to the composite with 30 wt % layer aligned BNNs increased more quickly than that attached to pure PFA, and the maximum temperate was significantly reduced by 13 °C. This result demonstrates that the alignment of BNNs in PFA composite improves the heat dissipation capability of composite.

### 3.5. Dielectric Properties of the Composite

Apart from a high TC, electronic packaging materials also need a low dielectric constant and dielectric loss. Generally, a low dielectric constant can reduce the time of signal propagation in electronic devices, while a low dielectric loss tangent can decrease heat production of dielectric materials under electric field. Figure 7a,b depicts the variation of dielectric constant and dielectric loss tangent as a function of frequency for pure PFA and the BNNs–PFA composites. The dielectric constants of pure PFA and composites all had excellent frequency stability. However, the dielectric constants of these PFA composites increased slightly with the increasing BNNs content. This is because the increasing BNNs content introduce more charge carriers [12,38]. Nevertheless, the dielectric constants for all the composites were maintained even at a low level. In particular, the 30 wt % BNNs–PFA composite had a dielectric constant of 3.57 at frequency of 10^3^ Hz, which increased slightly in comparison with that of pure PFA (~2.90).

Figure 7b indicates a decreasing dielectric loss tangent of the composites with the increasing frequency. This may be because the interfacial polarization requires enough time and cannot catch up with external electric field at high frequency [11,38]. Nevertheless, we found that the dielectric loss tangent of composites was similar to that of pure PFA. These results demonstrated that adding BNNs into the PFA matrix had little influence on the dielectric constant and dielectric loss of composites. Overall, the results of this study demonstrate that the BNNs–PFA composites not only have high anisotropic TC, but also the low dielectric property.

## 4. Conclusions

In this work, we fabricated anisotropic thermally conductive PFA composite with low dielectric constant. We found that the BNNs–PFA composites possessed good thermal stability. When the BNNs content exceeded 10 wt %, the BNNs were well layer aligned in the PFA matrix owing to hot pressing, and the fabricated composites exhibited obvious anisotropic TC. The in-plane TC and through-plane TC of PFA composite with 30 wt % BNNs were 4.65 and 1.94 W·m^−1^·K^−1^, respectively. The TC can be well simulated by an EMT model. In addition, the composites possessed a low dielectric constant and dielectric loss tangent. Considering that hot pressing is facile and of low cost, our approach is effective for fabricating anisotropic thermally conductive composite with low dielectric constant, which can be used in electronic packaging.

## Figures and Tables

**Figure 1 polymers-11-01638-f001:**
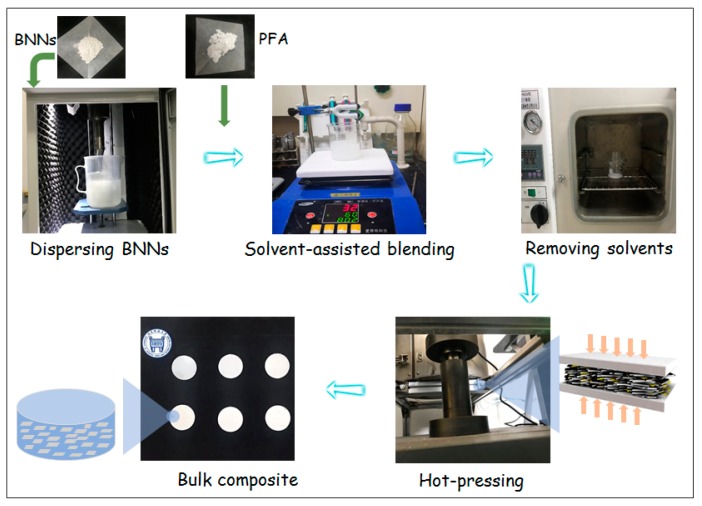
Schematic of the fabrication process for the perfluoroalkoxy (PFA) composites with layer-aligned boron nitride nanosheets (BNNs). First, the produced BNNs were dispersed in solvents via sonication. Then, PFA powder was added into BNNs dispersion and mixed using a heating magnetic stirrer while removing the solvent. Afterwards, the BNNs–PFA slurry was dried in a vacuum oven to remove the residual solvent. Finally, the obtained powder was put into a mold and hot-pressed into bulk composites.

**Figure 2 polymers-11-01638-f002:**
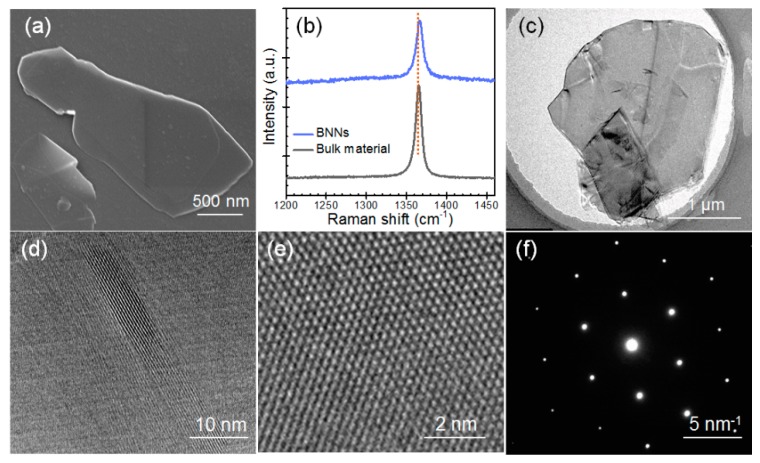
SEM image (**a**), Raman spectra (**b**), Transmission Electron Microscopy (TEM) image (**c**), high resolution TEM (**d**,**e**), and electron diffraction pattern (**f**) of the BNNs.

**Figure 3 polymers-11-01638-f003:**
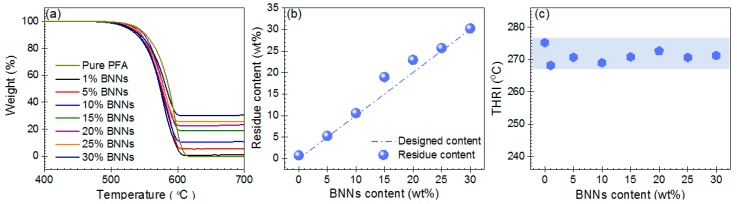
(**a**) Thermogravimetric analysis curves for pure PFA and the BNNs–PFA composites. (**b**) The relationship between the initial designed and residual BNNs content obtained in thermogravimetric analysis. (**c**) The heat resistance index (THRI) for pure PFA and the BNNs–PFA composites. THRI = 0.49[*T*_5_ + 0.6(*T*_30_ – *T*_5_)], where *T*_5_ and *T*_30_ is the temperature at 5 and 30 % weight loss in (**a**), respectively.

**Figure 4 polymers-11-01638-f004:**
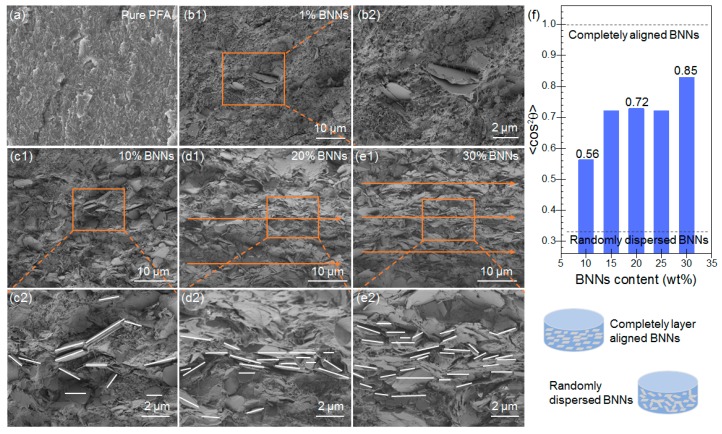
SEM images of the fractured surfaces for pure PFA (**a**), PFA composites containing 1 wt % (**b1**,**b2**), 10 wt % (**c1**,**c2**), 20 wt % (**d1**,**d2**) and 30 wt % (**e1**,**e2**) BNNs. (**f**) The <cos^2^*θ*> determined based on the image analysis of frequency versus *θ* from the SEM images of PFA composites containing 10, 15, 20, 25, and 30 wt % BNNs.

**Figure 5 polymers-11-01638-f005:**
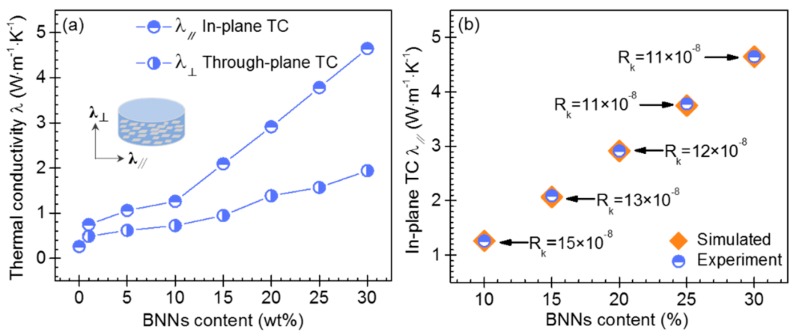
(**a**) In-plane TC (*λ*_‖_) and through-plane TC (*λ*_⊥_) of the PFA composites with layer aligned BNNs. (**b**) Extraction of *R_k_* values for BNNs–PFA composites based on the EMT model. The unit of the *R_k_* is m^2^·K·W^−1^.

**Figure 6 polymers-11-01638-f006:**
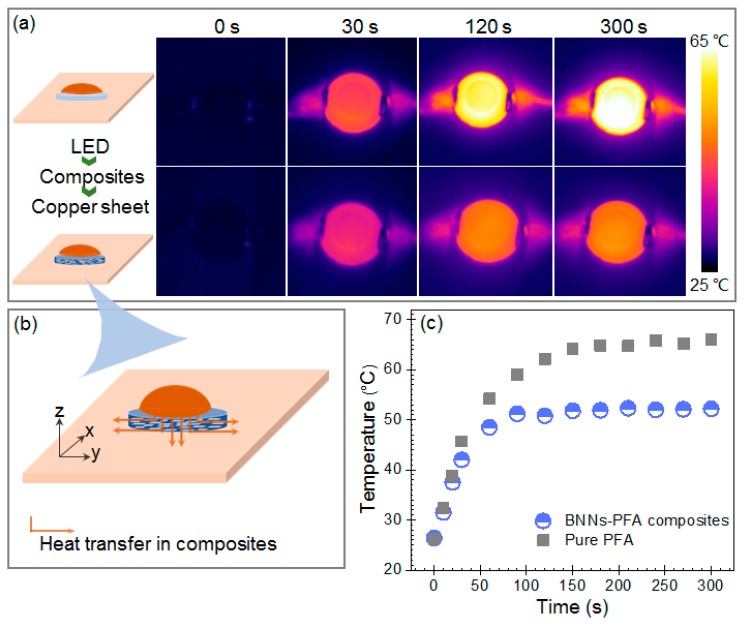
(**a**) Infrared thermal images for the LED lamps attached to pure PFA and the PFA composites with 30 wt % layer aligned BNNs. In the experiment, the back surface of a high-power LED lamp was attached to the composite. Then, a thin copper sheet was connected to the bottom side of pure PFA and BNNs–PFA composite, as shown by the insets. (**b**) Schematic of the heat conduction in PFA composites with layer aligned BNNs. (**c**) Surface temperatures as a function of time for the LED lamps attached to pure PFA and the PFA composites with 30 wt % layer aligned BNNs.

**Figure 7 polymers-11-01638-f007:**
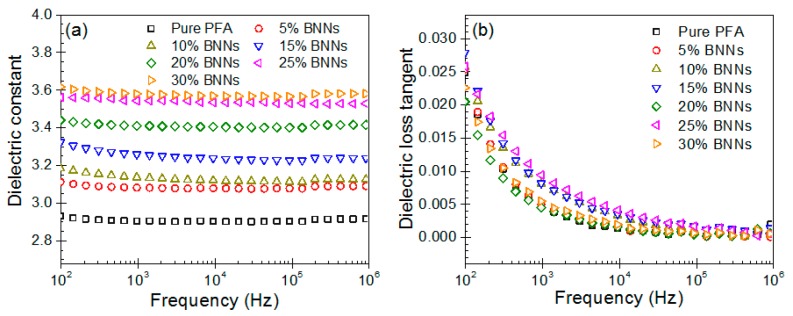
Dielectric constant (**a**) and dielectric loss tangent (**b**) as a function of frequency for pure PFA and the BNNs–PFA composites.

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
