# Peer review of "Anisotropic Thermally Conductive Perfluoroalkoxy Composite with Low Dielectric Constant Fabricated by Aligning Boron Nitride Nanosheets via Hot Pressing"

_polymers, 2019, doi:10.3390/polym11101638_

Round 1

Reviewer 1 Report

This is an interesting work which described a simple and low cost method (hot pressing) of preparation of thermally conducive PFA/boron nitride nanosheets composites. Results are interesting. In my view this may be considered for publication after the following modifications:

1.Authors should provide more information about used BN.

2.The BNNs were prepared by liquid phase exfoliation method. The BN-ethanol-water mixture was sonicated using a tip sonicator for 180 min. Why time of 180 min was used?. Maybe, shorter time (for example of 60 min) will be enough for exfoliation of BN. The solvent plays also an important role in aiding exfoliation. Why authors have used ethanol-water mixture. Have the authors checked that prolonging the exfoliation time can cause damage of BN sheets ? How was checked the effectiveness of the used combination of sonication and centrifugation method in separating BN to single or several nanometer thickness? Authors should provide such information in introduction or in experimental part.

3.Have the authors checked that prolonging the exfoliation time of can cause damage of BN sheets, which can influence on their thermal conductivity? For carbon nanotubes it was established that due to the sonication treatment, progressive damage to the CNTs' graphitic structure occurs, with the formation of structural defects on the CNT surface, the concentration of which rises with the increasing sonication time.

4. Please also report references reported according to journal style.

Reviewer 2 Report

Dear Editor,

Thank you for trusting me and offering revising the work by Xinru Zhang and colleagues.

The manuscript entitled “Anisotropic thermally conductive perfluoroalkoxy composite with low dielectric constant fabricated by aligning boron nitride nanosheets via hot pressing” presents interesting results and I reckon it fits well with the scope of the Polymers journal. In the paper the authors describe the approach for effective fabrication of anisotropic thermally conductive composites with low dielectric constant. Such composites can be potentially used in electronic packaging.
I found the manuscript to be written understandably, though I would suggest two improvements:

The abstract should be re-written so that it is clear what the motivation for conducting the research is and what the scientific hypotheses are. I suggest the following construction:
(i) Define a problem using in simple language so it is easy to understand by the general reader of Polymers journal,
(ii) Write why it is important to solve this problem, and
(iii) How you solve the problem, i.e., what is the scientific approach to solve the problem, and
(iv) what the main results of the research are, and how they impact the research field.

And again, just like in the abstract, in the Introduction it is not easy for me to find the scientific hypotheses and problems this research is solving. I suggest to make the Introduction more coherent and concrete.

The Figures and their descriptions are very clear for me. I did not find any major problem regarding the scientific content.
